# LYAPUNOV STABILITY LEARNING WITH NONLINEAR CONTROL VIA INDUCTIVE BIASES

## ABSTRACT

Finding a control Lyapunov function (CLF) in a dynamical system with a controller is an effective way to guarantee stability, which is a crucial issue in safety-concerned applications. Recently, deep learning models representing CLFs have been applied into a learner-verifier framework to identify satisfiable candidates. However, the learner treats Lyapunov conditions as complex constraints for optimisation, which is hard to achieve global convergence. It is also too complicated to implement these Lyapunov conditions for verification. To improve this framework, we treat Lyapunov conditions as inductive biases and design a neural CLF and a CLF-based controller guided by this knowledge. This design enables a stable optimisation process with limited constraints, and allows end-to-end learning of both the CLF and the controller. Our approach achieves higher convergence rate and larger region of attraction (ROA) in learning the CLF compared to existing methods among abundant experiment cases. We also thoroughly reveal why the success rate decreases with previous methods during learning.

## 1 INTRODUCTION

In recent years, deep learning models have demonstrated significant potential in controlling nonlinear dynamical systems with high performance results within various applications such as car navigation (Han et al., 2022), robot scooping (Niu et al., 2023), and quadrotor flying (O'Connell et al., 2022). However, as deep learning methods led to black-box models, when safety concerns cannot be overlooked, the real-life implementation requires theoretical analysis of the stability performance, creating an open question (Amodei et al., 2016).

For a dynamical system with a controller, the theory of Lyapunov stability is an important approach in stability analysis, which determines a control Lyapunov function (CLF) and a region of attraction (ROA) (Giesl & Hafstein, 2015). Within the ROA, the system is proven to be asymptotically stable around the equilibrium. Nonetheless, this theory does not provide a way to systematically discover a CLF. Designing a feasible framework to achieve this goal becomes a necessary research topic.

Previous works focus on finding CLFs for polynomial dynamical systems (Khodadadi et al., 2014; Majumdar et al., 2013). For nonlinear dynamics, a learner-verifier framework is proposed with polynomial CLF candidates to guarantee stability (Ravanbakhsh & Sankaranarayanan, 2019), which is extended to learn neural CLFs with a linear-quadratic regulator (LQR) controller (Chang et al., 2019). Additionally, a mathematical analysis is provided (Zhou et al., 2022) and learning piecewise models as CLFs and dynamics is studied for simplicity (Dai et al., 2020).

However, three shortcomings exist within this learner-verifier framework. First, three Lyapunov conditions are converted to soft constraints in the trainer (Chang et al., 2019; Dai et al., 2020; Zhou et al., 2022), but the complex combination of loss weights makes it hard to optimise the neural CLF candidate to satisfy all these conditions at the same time.

Second, to release this difficulty in learning, the overlooked region around the equilibrium is introduced (Chang et al., 2019; Zhou et al., 2022; Zinage & Bakolas, 2023). The verifier will not check the satisfaction of Lyapunov conditions within this region:

$$\Omega_0 = \{\mathbf{s} \mid \|\mathbf{s} - \mathbf{s}^*\| < r_0\}, \tag{1}$$

where $\mathbf{s}^*$ is the equilibrium state and $r_0 \in \mathbb{R}_+$ is a radius. But the existence of this region will lead to rough or even fake results deceiving the verifier, along with a controller failing to stabilise the system.

Table 1: Framework structures comparison of different methods

| METHODS | MODELS | EFFECTS FOR THE TRAINER AND THE VERIFIER | | IMPLEMENTATION | RESULTS |
|---|---|---|---|---|---|
| ULC (Zhou et al., 2022) NLC (Chang et al., 2019) | Vanilla neural networks | Multiple Lyapunov risk losses | Independent library, with overlooked region | Simple and fast | Inaccurate |
| Ours | Lyapunov inductive networks | Only one Lyapunov risk loss | Verified within trainer, no overlooked region | ↕ | ↕ |
| LSNNC (Dai et al., 2021) | Piecewise linear networks | Multiple Lyapunov risk losses | Nonlinear solver using CPU only | Complex and slow | Accurate |

Third, the verifier consists of a non-convex solver to check the Lyapunov conditions, which is independent from the deep learning library and usually runs on CPU only. To express Lyapunov conditions in the verifier, users have to extract weights from the neural CLF candidate and then hard-code this deep learning model, its derivative, the dynamics and controller models with only basic mathematical operations. The implementation will be impractical when models become complex.

To address these issues mentioned above, we propose an end-to-end pipeline for learning the neural CLF and controller for the nonlinear dynamical system. Apart from optimising constraints to meet the Lyapunov conditions, we first treat Lyapunov conditions as inductive biases. As compared in Table 1, by incorporating this knowledge into the neural CLF and the controller, we can satisfy mostly the Lyapunov conditions before learning, reducing the optimisation burden and improving the possibility of convergence. Additionally, we can perform the verification step using only the deep learning library, simplifying the framework for the easy implementation.

**Contribution**. First, we design the neural CLF and the nonlinear controller with one neural network instructed by Lyapunov conditions as prior inductive biases, which enables us to train them in a synthesis way. Second, we incorporate the verifier into trainer and propose an end-to-end framework to find satisfiable CLFs for nonlinear dynamical systems. Third, we provide a thorough investigation with regard to the robustness of different methods, the reasons of their differences in performance, and the effects of the other heuristic learning factor called the geometric shaping loss. We demonstrate robust convergence and superior performance in finding CLFs with larger ROAs compared to counterparts.

## 2 RELATED WORKS

**Lyapunov theory with classical methods.** The application of the Lyapunov theory ranges from simple to complex cases. For linear dynamical systems, the linear-quadratic (LQR) controller is able to achieve the Lyapunov stability with a quadratic Lyapunov function (Khalil, 2015). The control-affine polynomial dynamics is also proven to be stable with a polynomial controllers and a sum-of-square (SOS) polynomial Lyapunov function using semidefinite programming (SDP) (Majumdar et al., 2013; Khodadadi et al., 2014; Jarvis-Wloszek et al., 2003; Henrion & Garulli, 2005). Besides, the SDP optimisation is widely expanded in a rich group of work (Parrilo, 2000; Papachristodoulou & Prajna, 2005). However, the scale of the SDP grows exponentially related to the dimension of dynamics with a SOS low-degree polynomial Lyapunov function (Parrilo, 2000).

**Lyapunov theory for stability estimation and analysis.** For nonlinear systems without controller, the research purpose is to prove the feature of stability. Researchers firstly turn to deep learning to directly approximate known regions of attractions (ROAs) (Berkenkamp et al., 2016a; Richards et al., 2018). To find a Lyapunov function to prove asymptotic stability, the learner-verifier framework proposed in (Ravanbakhsh & Sankaranarayanan, 2019) has been further extended with deep learning (Grüne, 2020; Abate et al., 2020; Dai et al., 2020; Grüne, 2021; Gaby et al., 2022). To approximate the Lyapunov function and verify satisfaction of the Lyapunov conditions, deep learning models have been applied with a SMT solver (Grüne, 2020; Abate et al., 2020). Piecewise linear systems have also been addressed using ReLU-activated networks with the MIP solver (Dai et al., 2020). Convergence has been analytically studied and networks have been designed to accelerate convergence (Grüne, 2021; Gaby et al., 2022). Despite the increased expression ability in deep learning for approximating the Lyapunov function, verifying the Lyapunov conditions among the whole defined region within non-convex solvers is difficult to implement. The framework is also unstable with complex optimisation targets. Moreover, the MIP solver requires piecewise linear functions, which limits its usage when other known or learnt nonlinear dynamics models are available.

**Lyapunov theory for control with stability guarantee.** Gaussian process models are firstly applied to improve the accuracy of dynamical models for more reliable control (Berkenkamp & Schoellig, 2015; Sui et al., 2015; Berkenkamp et al., 2016b; Koller et al., 2018), as well as for learning safe

regions for control policy (Schreiter et al., 2015; Turchetta et al., 2019). Recent, several works focus on introducing a controller that can be proven stable with respect to Lyapunov theory (Chang et al., 2019; Dai et al., 2021; Zhou et al., 2022; Zinage & Bakolas, 2023). In (Chang et al., 2019), the learner-verifier framework is first used to find a CLF candidate for nonlinear dynamical systems using the LQR controller, and this methodology is later extended to nonlinear controllers (Dai et al., 2021; Zhou et al., 2022). This framework is combined with Koopman operator theory to learn the dynamical system at the same time (Zinage & Bakolas, 2023). Additionally, a method is proposed for decomposing the dynamical system into a continuous branch of Lyapunov functions and controllers for subsystems for generalisation (Zhang et al., 2023). However, as noted earlier, challenges exist in synthetically learning controllers and the corresponding CLF with a complex optimisation target, where careful tuning weights of the loss for convergence is required.

## 3 PRELIMINARIES

We consider the nonlinear control-affine Lipschitz continuous dynamical systems subject to bounded control inputs:

$$\dot{\mathbf{s}} = \mathbf{f}(\mathbf{s}) + \mathbf{g}(\mathbf{s})\mathbf{u}(\mathbf{s}), \tag{2}$$

where $\Omega$ is a closed state domain, $\mathbf{s} \in \Omega \subseteq \mathbb{R}^n$, $\mathbf{f} : \Omega \to \mathbb{R}^n$, $\mathbf{g} : \Omega \to \mathbb{R}^{n \times m}$ and $\mathbf{u} : \Omega \to [\mathbf{u}_{\min}, \mathbf{u}_{\max}] \subseteq \mathbb{R}^m$. The bounded controller corresponds to real-world situations, following previous works (Zhou et al., 2022; Dai et al., 2021)

**Lyapunov conditions.** Given a goal equilibrium state $\mathbf{s}^*$ with control $\mathbf{u}^* = \mathbf{u}(\mathbf{s}^*)$, the control Lyapunov function (CLF) $V(\mathbf{s})$ satisfies the following conditions (Giesl & Hafstein, 2015):

$$\dot{V}(\mathbf{s}) < 0 \ \forall \mathbf{s} \in \Omega, \ \mathbf{s} \neq \mathbf{s}^*, \tag{3a}$$

$$V(\mathbf{s}) > 0 \ \forall \mathbf{s} \in \Omega, \ \mathbf{s} \neq \mathbf{s}^*, \tag{3b}$$

$$V(\mathbf{s}^*) = 0 \ , \ \dot{V}(\mathbf{s}^*) = 0, \tag{3c}$$

where $\dot{V}(\mathbf{s}) = \nabla V(\mathbf{s})^{\mathrm{T}} \dot{\mathbf{s}} = \nabla V(\mathbf{s})^{\mathrm{T}} (\mathbf{f} + \mathbf{g}\mathbf{u})$ is the time derivative, and $\nabla V(\mathbf{s}) = \frac{\partial V(\mathbf{s})}{\partial \mathbf{s}}$ is the partial derivative with respect to $\mathbf{s}$.

**Region of Attraction (ROA).** We consider the compact region defined by $\mathcal{D}_\rho = \{\mathbf{s} \mid V(\mathbf{s}) \leq \rho, \rho \in \mathcal{R}_+\} \subseteq \Omega$, where all Lyapunov conditions in Equation (3) are met. The largest level set is defined by $\mathcal{D} = \max_\rho \mathcal{D}_\rho$ as the ROA. Here, the CLF proves the asymptotic stability of the system within ROA around the equilibrium. As depicted in Figure 1, as long as the initial state $\mathbf{s}_0$ is within the corresponding ROA, because $\dot{V}(\mathbf{s}) < 0$, the CLF value strictly decreases to 0 with time according to the Lyapunov conditions, which implies that the state will converge to $\mathbf{s}^*$:

$$\forall \mathbf{s}(0) = \mathbf{s}_0 \in \mathcal{D} : \lim_{t \to \infty} V(\mathbf{s}(t)) \to 0 \Rightarrow \lim_{t \to \infty} \mathbf{s}(t) \to \mathbf{s}^*.$$

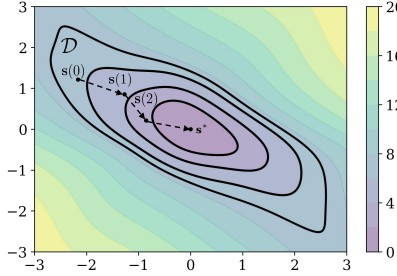

Figure 1: An illustration of a CLF candidate satisfying Lyapunov conditions within the closed state region $\Omega$. It displays the maximum level set $\mathcal{D}$ that can be found. The CLF value is strictly decreasing with respect to time in $\mathcal{D}$ because it is a closed region.

To the best of our knowledge, there is no way to design a general form for the CLF and the bounded controller to completely satisfy all Lyapunov conditions. Utilising deep learning models to represent the CLF is a feasible way to find a suitable CLF. However, training vanilla neural networks with excessive constraints shown in Equation (3) is difficult because the searching space of weights to

balance constraints is too broad. If we consider the knowledge as inductive biases and design the neural CLF candidate and controller instructed by them, we can simplify the learning framework, release the training burden, and still retain theoretical guarantees presented in Zhou et al. (2022).

## 4 METHOD

In this section, we update the learner-verifier framework with an end-to-end learning pipeline illustrated in Algorithm 1 to find the CLF and nonlinear controller for the dynamical system, which comprises two parts: pretraining the dynamics and training the entire neural CLF control system.

### 4.1 DYNAMICAL SYSTEM PRETRAINING

When dynamics is unknown or simplified dynamics is needed, it can be approximated simply using regression before learning the neural CLF. Two multilayer perceptrons (MLPs) $\mathbf{f}_{\phi_1}$ and $\mathbf{g}_{\phi_2}$ can be used to model $\mathbf{f}$ and $\mathbf{g}$, respectively, with $\phi_1$ and $\phi_2$ being the parameters. Given samples $(\mathbf{s}, \mathbf{u}, \dot{\mathbf{s}})$, the $L^2$ norm is used for the optimisation: $\mathcal{L}_{\mathrm{dyn}}(\mathbf{s}) = \|\dot{\mathbf{s}} - \hat{\dot{\mathbf{s}}}\|_2$, where $\hat{\dot{\mathbf{s}}} = \mathbf{f}_{\phi_1}(\mathbf{s}) + \mathbf{g}_{\phi_2}(\mathbf{s})\mathbf{u}$ and $\dot{\mathbf{s}}$ represent the time derivative of the state obtained from the network and the ground truth, respectively.

Following the proof presented in (Zhou et al., 2022), the effect of the difference of the learned dynamics and the ground truth can be eliminated by introducing a positive number $b$ in condition (3a) for training: $\dot{V}(\mathbf{s}) < -b$. Here $b \geq M(K_{\mathrm{dyn}}\delta + \epsilon + K_\phi\delta) > 0$ is related to the norm of the derivative of the Lyapunov function $M = \|\frac{\partial V}{\partial \mathbf{s}}\|$, the learnt error $\epsilon$, the sample interval $\delta$, the Lipschitz constants of the dynamics $K_{\mathrm{dyn}}$ and the learnt dynamics $K_\phi$. Still, the determination of $b$ requires some prior knowledge. For simplicity, we utilise $\mathbf{f}$ and $\mathbf{g}$ to refer the dynamics afterwards.

### 4.2 SELF-SUPERVISED TRAINING

The self-supervised learning pipeline is shown in Figure 2. To reduce the complexity of the optimisation target, we carefully design the neural CLF candidate and the controller to satisfy most of the constraints of Equation (3). Next, we introduce the optimisation process.

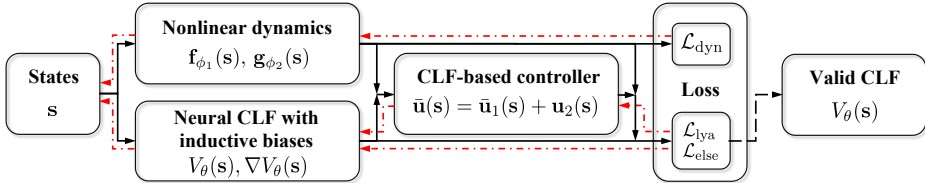

Figure 2: The self-supervised framework to synthetically learn the neural CLF and the CLF-based controller for nonlinear dynamics. Solid black arrows represent forward flows: Given the state $\mathbf{s}$, the values of dynamics $\mathbf{f}, \mathbf{g}$ and Lyapunov function $V, \nabla V$ are obtained and then fed into the CLF-based control policy for control $\mathbf{u}$. Then the loss is calculated for optimisation. Red dash-dot arrows represent backpropagation flows for different loss terms.

**Control Lyapunov Function Candidate.** With proper design of the architecture of the CLF candidate, we can meet the semi-definite positive Lyapunov condition around the equilibrium $\mathbf{s}^*$. The sum-of-squares (SOS) structure is utilised for polynomial Lyapunov functions (Papachristodoulou & Prajna, 2005) and SOS neural Lyapunov functions are only applied to approximate known ROAs for classification (Berkenkamp et al., 2016a). Here, we build the sum-of-squares neural CLF candidate $V_\theta$ with network $\phi_\theta : \mathbb{R}^n \to \mathbb{R}^l$ as:

$$V_\theta(\mathbf{s}) = V_1(\mathbf{s}) + V_2(\mathbf{s}) = \|\phi_\theta(\mathbf{s}) - \phi_\theta(\mathbf{s}^*)\|_2^2 + k\log\left(1 + \left(\sum_i^n (s_i - s_i^*)\right)^2\right), \quad (4)$$

where $\|\cdot\|_p$ denotes the $L^p$ norm, $k > 0$ is a hyperparameter, $l$ is the hidden dimension, and $\theta$ is the network parameter. It can be easily verified that this candidate satisfies $V_\theta(\mathbf{s}^*) = 0$, $V_\theta(\mathbf{s}) > 0, \mathbf{s} \neq \mathbf{s}^*$, and $\dot{V}_\theta(\mathbf{s}^*) = \nabla V_\theta(\mathbf{s}^*)^{\mathrm{T}}\dot{\mathbf{s}}^* = \nabla V_\theta(\mathbf{s}^*)^{\mathrm{T}}\mathbf{0} = 0$. Therefore, conditions (3b) and (3c) are satisfied.

The introduction of the second loss term $V_2$ can serve as a valuable augmentation for $V_\theta$ because it guarantees that $V_\theta$ equals $0$ only when the state is at the equilibrium point $\mathbf{s}^*$. Its gradient is $\nabla V_2(\mathbf{s}) = \frac{2\sum_i^n (s_i - s_i^*)}{1 + (\sum_i^n (s_i - s_i^*))^2}$, whereby $\lim_{\|\mathbf{s} - \mathbf{s}^*\|_2 \to 0} \nabla V_2(\mathbf{s}) = 0$ and $\lim_{\|\mathbf{s} - \mathbf{s}^*\|_2 \to \infty} \nabla V_2(\mathbf{s}) = 0$. This indicates that $V_2(\mathbf{s})$ increases gradually and smoothly when close or far from the equilibrium point and does not overpower the expression of the neural network part. So we can highlight the effect of $V_1(\mathbf{s})$ during training.

**CLF-Based Controller.** Given a CLF, the Sontag's feedback control (Sontag, 1989; Freeman & Primbs, 1996) can ensure the condition (3a) is met. However, the controlling output can be excessively large, which is unrealistic in the real world. Under the constraint of the bounded control output, we build the controller model as follows:

$$\mathbf{u}(\mathbf{s}) = \bar{\mathbf{u}}_1(\mathbf{s}) + \mathbf{u}_2(\mathbf{s}) = \mathrm{clip}(\mathbf{u}_1(\mathbf{s}), \mathbf{u}_{1,\min}, \mathbf{u}_{1,\max}) + \mathbf{u}_2(\mathbf{s}), \tag{5a}$$

$$\mathbf{u}_1(\mathbf{s}) = -\frac{\nabla V(\mathbf{s})^\mathrm{T} \mathbf{f} \mathbf{g}^\mathrm{T} \nabla V(\mathbf{s})}{\nabla V(\mathbf{s})^\mathrm{T} \mathbf{g} \mathbf{g}^\mathrm{T} \nabla V(\mathbf{s})}, \quad \mathbf{u}_2(\mathbf{s}) = -\mathbf{q}_1 \cdot \tanh(\mathbf{q}_2 \cdot \mathbf{g}^\mathrm{T} \nabla V(\mathbf{s})), \tag{5b}$$

where $\mathbf{q}_1, \mathbf{q}_2 \in \mathbb{R}_+^m$ are vectors with positive elements and $\cdot$ denotes the dot product for vectors. $\mathbf{u}_1$ aims to counterbalance the effect of the dynamical function $\mathbf{f}$. The function $\mathrm{clip}(\mathbf{x}, \mathbf{x}_{\min}, \mathbf{x}_{\max}) = \max(\min(\mathbf{x}, \mathbf{x}_{\max}), \mathbf{x}_{\min})$ constrains the values outside the range $[\mathbf{x}_{\min}, \mathbf{x}_{\max}]$ to the range edges, $\mathbf{u}_{1,\min} = \mathbf{u}_{\min} + \mathbf{q}_1$ and $\mathbf{u}_{1,\max} = \mathbf{u}_{\max} - \mathbf{q}_1$ are the control range for $\mathbf{u}_1$. $\mathbf{u}_2$ is already bounded by the $\tanh$ function and can drive the system to equilibrium, where $\mathbf{q}_1$ defines the amplitude of $\mathbf{u}_2$ and $\mathbf{q}_2$ controls how fast $\mathbf{u}_2$ reaches the maximum output.

Here we design a controller integrated with the CLF to approach condition (3a) without introducing a new neural network. For a rough estimation of the time derivative of CLF, given Equation (2) and (5b), when $\mathbf{u}_1(\mathbf{s}) = \bar{\mathbf{u}}_1(\mathbf{s})$, we have

$$\dot{V}(\mathbf{s}) = \nabla V^\mathrm{T} \dot{\mathbf{s}} = \nabla V^\mathrm{T}(\mathbf{f} + \mathbf{g}\mathbf{u}_1 + \mathbf{g}\mathbf{u}_2)) \text{ use Eq.(2)}$$

$$= \nabla V^\mathrm{T} \mathbf{f} - \frac{\nabla V^\mathrm{T} \mathbf{g} \nabla V^\mathrm{T} \mathbf{f} \mathbf{g}^\mathrm{T} \nabla V}{\nabla V^\mathrm{T} \mathbf{g} \mathbf{g}^\mathrm{T} \nabla V} - (\mathbf{g}^\mathrm{T} \nabla V)^\mathrm{T}(\mathbf{q}_1 \cdot \tanh(\mathbf{q}_2 \cdot \mathbf{g}^\mathrm{T} \nabla V)) \text{ use Eq.}(5b) \tag{6}$$

$$= -(\mathbf{g}^\mathrm{T} \nabla V)^\mathrm{T}(\mathbf{q}_1 \cdot \tanh(\mathbf{q}_2 \cdot \mathbf{g}^\mathrm{T} \nabla V)).$$

Since the signs of the corresponding elements between $\mathbf{g}^\mathrm{T} \nabla V$ and $\tanh(\mathbf{g}^\mathrm{T} \nabla V)$ are always the same and $(\mathbf{q}_1, \mathbf{q}_2)$ are positive vectors, $(\mathbf{g}^\mathrm{T} \nabla V)^\mathrm{T}(\mathbf{q}_1 \cdot \tanh(\mathbf{q}_2 \cdot \mathbf{g}^\mathrm{T} \nabla V))$ is positive, i.e., $\dot{V}(\mathbf{s}) \leq 0$.

As discussed in section 3, the whole satisfaction within region $\mathbf{u}_1(\mathbf{s}) \neq \bar{\mathbf{u}}_1(\mathbf{s})$ can be approached by optimisation, with only one Lyapunov risk loss term shown in Equation (7b).

**Optimisation.** The training of the neural CLF and the CLF-based controller in Equation (4) and (5) relies on self-supervised learning. We define the loss function as:

$$\mathcal{L}(\mathbf{s}) = \mathcal{L}_{\mathrm{lya}}(\mathbf{s}) + \mathcal{L}_{\mathrm{else}}(\mathbf{s}), \tag{7a}$$

$$\mathcal{L}_{\mathrm{lya}}(\mathbf{s}) = \frac{1}{N} \sum_{i=1}^{N} \lambda_1 w(\mathbf{s}_i)(\max(\dot{V}_\theta(\mathbf{s}_i) + b, 0))^2 \tag{7b}$$

$$\mathcal{L}_{\mathrm{else}}(\mathbf{s}) = \frac{1}{N} \sum_{i=1}^{N} \lambda_2 w(\mathbf{s}_i) \|\mathbf{u}_1(\mathbf{s}_i) - \bar{\mathbf{u}}_1(\mathbf{s}_i)\|_2 , \tag{7c}$$

where $\dot{V}_\theta(\mathbf{s}_i) = \nabla \mathbf{V}_\theta(\mathbf{f} + \mathbf{g}\bar{\mathbf{u}})(\mathbf{s}_i)$, $\mathcal{L}_{\mathrm{lya}}(\mathbf{s})$ represents the Lyapunov risk for condition (3a), $\mathcal{L}_{\mathrm{else}}(\mathbf{s})$ enforces a soft constraint to prevent the controller output from exceeding the constraint range, $N$ is the number of samples, $w(\mathbf{s}_i)$ is the weight of different sample, and $(\lambda_1, \lambda_2, b)$ are hyperparameters with $\lambda_1 > 0, \lambda_2 > 0, b \geq 0$. $w(\mathbf{s}_i)$ can prioritise training on some target regions and is set as 1 for simplicity in our work. As $\mathcal{L}_{\mathrm{lya}}(\mathbf{s})$ decreases, a neural CLF and a controller meeting the satisfaction could be found during the self-supervised learning.

As for implementation and optimisation, four things are worth pointing out. First, the Lyapunov risk loss $\mathcal{L}_{\mathrm{lya}}(\mathbf{s})$ is simplified because we only need to satisfy condition (3a). In previous works, all three Lyapunov conditions in Equation (3) are converted to soft constraints as Lyapunov risks within the

---

**Algorithm 1:** Lyapunov-Stable Control

---

**Function** `TrainLyapunovSystem`$(\Omega, \mathbf{f}_{\phi_1}(\mathbf{s}), \mathbf{g}_{\phi_2}(\mathbf{s}))$

    Set CLF $V_\theta(\mathbf{s})$, optimiser `optim`, total iteration steps `max_step`;

    $\mathbf{S} \leftarrow$ `GenerateGridSamples`$(\Omega)$;    `/* Capital letter: batch. */`

    **for** $i = 1$ **to** *max_step* **do**

        $\nabla \mathbf{V}_\theta(\mathbf{S}) \leftarrow$ `autograd`$(\mathbf{V}_\theta(\mathbf{S}))$;

        $\mathbf{U}_1, \bar{\mathbf{U}}_1, \bar{\mathbf{U}} \leftarrow$ Equation (5a) and (5b);

        $\dot{\mathbf{S}} \leftarrow \mathbf{f}_{\phi_1}(\mathbf{S}) + \mathbf{g}_{\phi_2}(\mathbf{S})\bar{\mathbf{U}}$;

        $\mathcal{L} = \frac{1}{|\mathbf{S}|}\left(\lambda_1(\max(\nabla \mathbf{V}_\theta \dot{\mathbf{S}}, \mathbf{0}))^2 + \lambda_2\|\mathbf{U}_1 - \bar{\mathbf{U}}_1\|_2\right)$;

        $\mathcal{L}$.`backward()`;

        `optim.step()`;

        `satisfiable, counterexamples` $\leftarrow$ `CheckSatisfiability()`;

        $\mathbf{S} \leftarrow (\mathbf{S} \setminus$ `old counterexamples`$) \cup$ `counterexamples`;

        **if** $\mathcal{L}_{lya,\,verify} \rightarrow 0$ **then** break;

    **return** $V_\theta(\mathbf{s})$

**Function** `Main()`

    **Input:** verified region $\Omega$, controller parameters $\mathbf{u}_{1,\min}, \mathbf{u}_{1,\max}, \mathbf{q}_1, \mathbf{q}_2$;

    $\mathbf{f}_{\phi_1}(\mathbf{s}), \mathbf{g}_{\phi_2}(\mathbf{s}) \leftarrow$ `PretrainDynamicsModel`$(\Omega)$;

    $V_\theta(\mathbf{s}) \leftarrow$ `TrainLyapunovSystem`$(\Omega, \mathbf{f}_{\phi_1}(\mathbf{s}), \mathbf{g}_{\phi_2}(\mathbf{s}))$;

    ROA $\leftarrow$ `FindROA`$(\Omega, \mathbf{V}_\theta(\mathbf{s}))$;

---

loss function (Chang et al., 2019; Dai et al., 2020; Zhou et al., 2022; Zinage & Bakolas, 2023):

$$\bar{\mathcal{L}}_{\text{lya}} = \frac{1}{N}\sum_{i=1}^{N}\left(C_1\max(\dot{V}_\theta(\mathbf{s}_i), b_1) + C_2\max(-V_\theta(\mathbf{s}_i), b_2)\right) + C_3 V_\theta^2(\mathbf{0}) + C_4\left\|\frac{\partial V_\theta}{\partial \mathbf{s}}(\mathbf{0})\right\|, \quad (8)$$

where $C_1, C_2, C_3, C_4, b_1, b_2$ are weights. Concerning the high complexity of the combination of their loss weights, it is very difficult to tune these weights to achieve ideal results. Delicately tuned learning settings can also be unstable when dynamical systems change.

Second, the verification of condition (3a) can be simply accomplished using deep learning packages. Other non-convex solvers is not needed over the verification region to examine the satisfaction of multiple constraints. So the whole training process is end-to-end for easy implementation without hard-coding neural networks and related derivatives.

Third, there is no overlooked region around the equilibrium mentioned in Equation (1) to pass the verification process. The whole region will be utilised for exploration and verification, which will make sure the satisfaction region starts from the equilibrium. This is important for the robustness of the learning results.

Fourth, the loss in Equation (7) is only for successfully finding a satisfactory neural CLF. To encourage exploration for a larger ROA, other tuning terms such as the geometric shaping term can be useful:

$$\mathcal{L}_{\text{shape}} = \frac{1}{N}\sum_{i=1}^{N}\eta_1\left(\|\mathbf{H}\mathbf{s}_i\|_2^2 - \eta_2 V(\mathbf{s}_i)\right)^2, \quad (9)$$

where $\mathbf{H}$ is an orthogonal transformation matrix and $\eta_1, \eta_2 \in \mathbb{R}_+$ are two hyperparameters (Chang et al., 2019; Zhou et al., 2022). Equation (9) can help prevent the neural CLF from being flat and shape it to match the verified region $\Omega$. How this term works is also investigated in experiments.

## 5 EXPERIMENTS

### 5.1 ROBUSTNESS AND PERFORMANCE EVALUATION

The targets of this experiment are twofold. First, to compare the robustness of different methods, we examine their performance in finding neural CLF with related to a large amount of cases. Second, to

reveal the ability in finding large stability regions, we collect the areas of ROAs among those success cases for comparison.

**Experiment Setup.** Two nonlinear dynamical systems, unicycle path following and inverted pendulum, are used because their official implementations are available within previous works (Chang et al., 2019; Zhou et al., 2022; Dai et al., 2021; Zinage & Bakolas, 2023) and can be viewed as representative cases. We believe these two systems can serve as reliable indicators for evaluation.

The inverted pendulum with an unstable equilibrium $\mathbf{s}^* = (\theta^*, \omega^*) = (0, 0)$ is defined by:

$$\dot{\theta} = \omega, \ \dot{\omega} = (mgl \sin \theta - D\omega + u)/ml^2,$$

where the gravity and damping are $g = 9.81$ and $D = 0.1$, respectively. The learning region is defined as $\Omega = \{[\theta, \omega] | -4 \leq \theta \leq 4, -4 \leq \omega \leq 4\}$ with control limits $-u_{\mathrm{amp}} \leq u \leq u_{\mathrm{amp}}$, $u_{\mathrm{amp}} = 20$.

The wheeled vehicle path following problem is to control distance error $d_e$ and angle error $\theta_e$. The equilibrium is $\mathbf{s}^* = (d_e^*, \theta_e^*) = (0, 0)$ and the dynamical equation is:

$$\dot{d}_e = v \sin(\theta_e), \ \dot{\theta}_e = -(v \cos(\theta_e))/(1 - d_e \kappa) + u,$$

where $\kappa$ is the target path. The learning region is defined as $\Omega = \{[d_e, \theta_e] | -0.8 \leq d_e \leq 0.8, -0.8 \leq \theta_e \leq 0.8\}$, with control limits $-u_{\mathrm{amp}} \leq u \leq u_{\mathrm{amp}}$, $u_{\mathrm{amp}} = 5$.

For the first robustness target, we vary parameters of these dynamical systems to create sufficient cases shown in Table 2. This means that the learning procedure will repeat 150 and 90 times for each method to collect the success rate. In contrast, these systems were presented with only a set of fixed parameters before (Chang et al., 2019; Zhou et al., 2022; Dai et al., 2021; Zinage & Bakolas, 2023).

Table 2: Parameter settings for nonlinear dynamical systems

| EXPERIMENTS | PARAM1 | PARAM2 | SEEDS | TOTAL |
|---|---|---|---|---|
| INVERTED PENDULUM | $m : (0, 5, 0.8, 1.0, 1.2, 1.5)$ | $l : (0.8, 1.0, 1.2)$ | $11 - 20$ | $10 \times 5 \times 3 = 150$ cases |
| PATH FOLLOWING | $\kappa : (0.8, 1.0, 1.2)$ | $v : (1.0, 1.5, 2.0)$ | $11 - 20$ | $10 \times 3 \times 3 = 90$ cases |

For the second performance target, in each case where the learning process succeeds with a neural CLF candidate, we enumerate and find the closed contour $\overline{\mathcal{D}}$ covering the maximum region as the ROA, within which the Lyapunov conditions are verified. Then we calculate the area $A_{\mathcal{D}}$ within the contour using the Green's theorem: $A_{\mathcal{D}} = \iint_{\mathcal{D}} dx \, dy = \oint_{\overline{\mathcal{D}}} 0.5 \cdot (x \, dy - y \, dx)$.

Because the geometric shaping term presented in Equation (9) is utilised within previous works to improve performance (Chang et al., 2019; Zhou et al., 2022), we also evaluate its effects within the experiment. Here $\mathbf{H} = \mathbf{I}$ is the diagonal matrix because the learning region is square. We tune $\eta_2$ for each method, change $\eta_1$ from 0.1 to at most 9.0, and repeat the above whole testing procedure.

**Model Setup.** To benchmark our method, we employ three representative methods, the unknown Lyapunov control (ULC, Zhou et al. (2022)), neural Lyapunov control (NLC, Chang et al. (2019)) and the Lyapunov-stable neural network control (LSNNC, Dai et al. (2021)). NLC utilises a LQR controller and ULC utilises a LQR controller with a $\tanh$ function scaling the output range:

$$\begin{aligned} \mathbf{u} &= \mathbf{K}(\mathbf{s} - \mathbf{s}^*) + \mathbf{u}^*, \\ \mathbf{u} &= u_{\mathrm{amp}} \cdot \tanh(\mathbf{K}(\mathbf{s} - \mathbf{s}^*)) + \mathbf{u}^*. \end{aligned} \tag{10}$$

LSNNC utilises a neural network control $\mathbf{u} = \mathrm{clip}(\mathbf{u}_\phi(\mathbf{s}) - \mathbf{u}_\phi(\mathbf{s}^*) + \mathbf{u}^*, \mathbf{u}_{\mathrm{min}}, \mathbf{u}_{\mathrm{max}})$ and is considered as a more powerful method than NLC and ULC.

The sum-of-square (SOS) polynomial with the LQR controller serves as our baseline following previous works. For LQR, we first linearise the dynamical function at the equilibrium $\mathbf{s}^*$. If we can solve the related Riccati equation (Khalil, 2015) to obtain a positive semidefinite matrix $\mathbf{P}$, the quadratic form $V(\mathbf{s}) = \mathbf{s}^{\mathrm{T}} \mathbf{P} \mathbf{s}$ will be a Lyapunov function in the neighbourhood of the target point.

For NLC and ULC, the overlooked region $\Omega_0 = \{\mathbf{s} \,|\, \|\mathbf{s} - \mathbf{s}^*\| < r_0\}$ exists during the verification process, where $r_0 = 0.2$ and $0.1$ for the inverted pendulum and path following dynamics, respectively. Even though one result can pass the verification, it only proves that the controller can stabilise the system to the overlooked region, not to the equilibrium.

Table 3: Model and training settings of our method for two dynamical systems

| PARAMETERS | $\delta_{\text{learn}}$ | $\delta_{\text{verify}}$ | $k$ | $\mathbf{q}_1$ | $\mathbf{q}_2$ | $\lambda_1$ | $\lambda_2$ | $b$ | $\eta_1$ | $\eta_2$ | lr | $\triangle T$ |
|---|---|---|---|---|---|---|---|---|---|---|---|---|
| INVERTED PENDULUM | $2 \times 10^{-2}$ | $2 \times 10^{-3}$ | 6 | 2 | $10^3$ | 1 | 1 | 0.1 | 1 | 1 | 0.01 | 1 |
| PATH FOLLOWING | $5 \times 10^{-3}$ | $10^{-3}$ | 6 | 2 | $10^3$ | 1 | 1 | 0.1 | 2 | 0.2 | 0.01 | 2 |

For our model, we also test our neural Lyapunov function with the LQR controller (called ours-l) and the $\tanh$-LQR controller (called ours-t) shown in Equation (10). Considering the data efficiency, we optimise using relatively sparse grid samples (with interval $\delta_{\text{learn}}$) during training. For every $\triangle T$ epochs, we conduct the verification using dense grid samples (with interval $\delta_{\text{verify}}$), which returns counterexamples that violate the Lyapunov condition (3a) for training. The settings of our model are shown in Table 3.

**Results.** The learning results are compared in Table 4, where results with overlooked regions are underlined for distinction. Overall, our method outperforms counterparts with higher success rate and larger ROAs over successful learning cases.

For our framework, the high success rates imply that our method is more stable among different learning cases. This result supports our idea that designing the neural networks models instructed by Lyapunov conditions is better than only optimising them with constraints. After reducing the learning difficulties, we can retain a stable performance in success rate and then introduce more features to search for better results. Figure 3(a) presents two neural CLFs founded by our method.

The shaping term is beneficial for the optimisation, increasing the success rate and the area of ROA. This is because losses in Equation (7) are only for the success of learning, and the learner will concentrate on one success result each time. As revealed in Figure 3(b), the ROA can be small because of the shape of neural CLF. The introduce of this shaping term can help the optimisation path escape from local minima and encourage the learning process to search for a better solution.

Utilising different controllers also affects the learning results. The areas of ROA increase using our controller compared with the LQR-like controllers. This supports utilising an expressive controller which can better adjust for the satisfaction of the Lyapunov conditions. These LQR-like controllers cannot totally stabilise the path following system, so the overlooked region around the equilibrium is also needed for ours-t and ours-l.

For LSNNC, their performance is also comparative, achieving large area of ROA in the inverted pendulum system. But the success rate implies that the performance is not stable and the learning speed for complicated models with only CPU (8 hours at most for one case) is 20-100 times slower than other methods. Therefore, the learning test with the geometric shaping term is not considered, which can take several months to tune hyperparameters and examine thousands of cases. Concerning that the time consumption will increase greatly when systems become complex, we believe that designing an end-to-end pipeline for the acceleration on GPU is important.

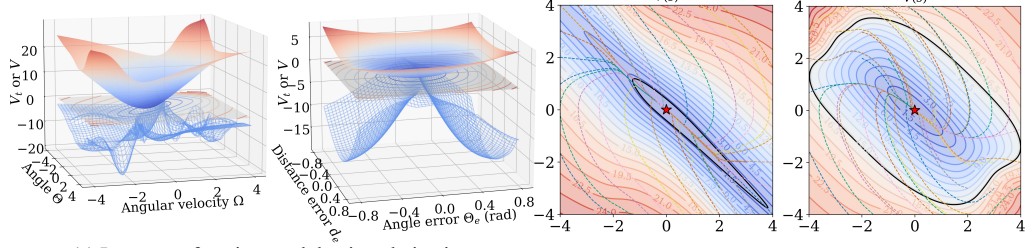

(a) Lyapunov functions and the time derivatitves    (b) ROAs obtained without/with the geometric shaping term

Figure 3: The training results of our method. (a) shows the learned neural CLF candidates (colormaps) and the derivatives (blue wireframes) for inverted pendulum dynamics (left) and path following (right) in 3D space. The contour of the CLF is also projected on the plane $V = 0$. (b) presents ROAs (black contours) of the CLFs. The area of ROA grows extensively larger with the geometric shaping term (right) than that without the term (left) because of a flatter shape. Colourful dash lines represent simulated trajectories, where black stars and red solid stars are end points and equilibriums, respectively. The deviation of stars represents how well the controller can stabilise the system.

Table 4: Success rate and areas of ROAs obtained by different methods[1]

| | | | | | INVERTED PENDULUM | | | |
|---|---|---|---|---|---|---|---|---|
| | | Ours | Ours-t | Ours-l | ULC (Zhou et al., 2022) | NLC (Chang et al., 2019) | LSNNC (Dai et al., 2021) | LQR |
| SUCCESS RATE | NO SHAPING | 82.67% | **96.67%** | 95.33% | 12% | 54.67% | 69.33% | - |
| | SHAPING | **97.33%** | 90% | 96.66% | 36.66% | 89.33% | - | - |
| AREA OF ROA | NO SHAPING | 6.90 ± 2.67 | 14.07 ± 3.84 | 5.47 ± 1.21 | 1.70 ± 1.25 | 6.84 ± 2.86 | **30.78 ± 8.27** | 6.86 |
| | SHAPING | **34.43 ± 4.53** | 8.53 ± 2.35 | 3.78 ± 0.98 | 11.33 ± 8.01 | 15.59 ± 4.16 | - | 6.86 |
| | | | | | PATH FOLLOWING | | | |
| | | Ours | Ours-t | Ours-l | ULC (Zhou et al., 2022) | NLC (Chang et al., 2019) | LSNNC (Dai et al., 2021) | LQR |
| SUCCESS RATE | NO SHAPING | **93.33%** | 90.8% | 88.89% | 44.44% | - | 28.89% | - |
| | SHAPING | **100%** | 100% | 87.78% | 43.33% | - | - | - |
| AREA OF ROA | NO SHAPING | **1.42 ± 0.41** | 0.88 ± 0.08 | 0.76 ± 0.17 | 0.79 ± 0.14 | FAILED | 0.81 ± 0.11 | FAILED |
| | SHAPING | **1.54 ± 0.37** | 1.03 ± 0.14 | 0.92 ± 0.28 | 0.93 ± 0.33 | FAILED | - | FAILED |

For the ULC and NLC, we believe that their low success rates should be blamed on the overlooked regions around the equilibrium set for the verifiers. Their successful outputs are also not meant to be the correct CLF and controller and we classify three representative types for their results:

1. Acceptable CLF: As shown in Figure 4(a), the Lyapunov conditions are almost meet, where only the minimum point slightly deviates from the target equilibrium.

2. Rough CLF: As shown in Figure 4(c-d), the Lyapunov conditions cannot be met around the target equilibrium, which is outlined by black dash-dot contours. If this conflicting region is small enough to locate within the overlooked region, the verification can be satisfied and the framework will output it as the result (Figure 4(c)). However, if the optimiser gets stuck within a local minima and cannot further shrink the conflicting region, the verifier will return false until the end of the learning (Figure 4(d)). The underlined result listed in Table 4 contains successful cases in this situation.

3. Fake CLF: As shown in Figure 4(b), the neural CLF candidate passes the verification but is not a CLF at all. The trainer finds a solution to deceive the verifier with the help of the overlooked region. Trajectories also show that the controller cannot stabilise the system to the equilibrium at all.

These examples explain why the overlooked region is not a wise option. The Lyapunov conditions around the equilibrium cannot be guaranteed, which can lead to the nonexistent of the ROA.

The investigation of the geometric shaping term is also presented within Figure 5. We can see that a suitable choice of weight $\eta_1$ can increase both the success rate and the area of ROA. But when $\eta_1$ grows larger, the success rate and even the area of ROA will decrease. This means that this term can help escape from local minimum points during optimisation, but overly emphasising this loss term will introduce more learning difficulties because it is not a requisite optimising target.

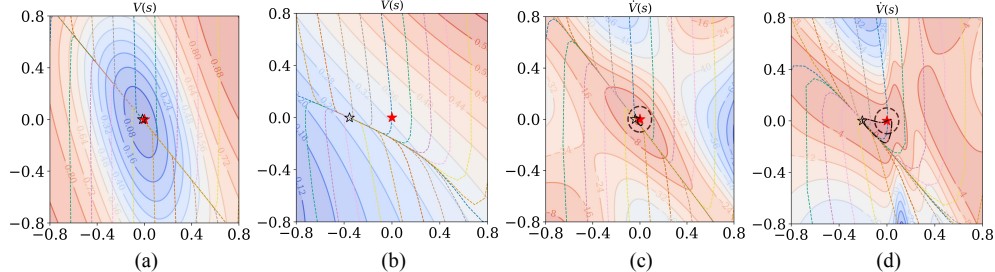

Figure 4: Four representative learning cases of NLC and ULC. (a-b) plot two neural CLF candidates $V(\mathbf{s})$ passing the verification, but (b) is **not** a CLF at all. (c-d) show the derivatives $\dot{V}(\mathbf{s})$, where the verification can pass in (c) but **cannot** in (d). Regions overlooked by the verifier ($\{\mathbf{s} \mid \|\mathbf{s}\| < 0.1\}$) and regions conflicting the Lyapunov condition ($\dot{V}(\mathbf{s}) > 0$) are outlined by dark red dash circles and black dash-dot contours, respectively. Colourful dash lines represent simulated trajectories, where black stars are end points. By contrast, the equilibrium points are marked with red solid stars.

## 5.2 EXTENSION

We set up our framework on two more complex dynamical systems for extension. One is the 4-DOF spacecraft rendezvous operation process given by the Hill Clohessy Wiltshire (HCW) equations:

$$\dot{x} = v_x, \ \dot{v}_x = 3n^2 x + 2n v_y + u_1, \ \dot{y} = v_y, \ \dot{v}_y = -2n v_y + u_2,$$

---

[1] The underlined results represent that the training of the corresponding methods can succeed only with the overlooked region $\Omega_0 = \{\mathbf{s} \mid \|\mathbf{s} - \mathbf{s}^*\| < r_0\}$.

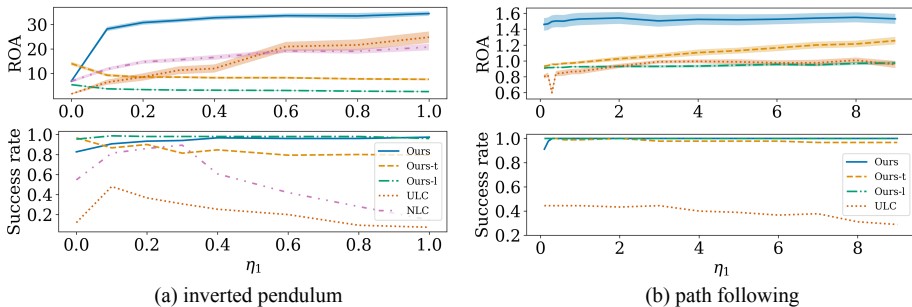

Figure 5: The area of ROA and success rate related to the value of $\eta_1$ in inverted pendulum (a) and path following dynamics (b). Increasing $\eta_1$ can enlarge the area of ROA, while can further decrease the success rate in training.

where $n = 1.1127 \times 10^{-3}$ is a variable related to the low earth orbit and the gravitation. The other is the 6-DOF 2D quadrotor model given by

$$\dot{x} = v_x, \ \dot{v}_x = -\sin(\theta)(u_1 + u_2)/m,$$
$$\dot{y} = v_y, \ \dot{v}_y = \cos(\theta)(u_1 + u_2)/m - g,$$
$$\dot{\theta} = w, \ \dot{w} = l(u_1 - u_2)/I,$$

where $(m, l, I, g) = (0.486, 0.25, 0.00383, 9.81)$ are the mass, length, inertia and gravity. The learnt CLF are shown in Figure 6(a), and we also sample 10 random initial states and plot the change of the CLF value among the simulated trajectories in Figure 6(b). It can be seen that as the states reach the equilibrium, the value decrease monotonically.

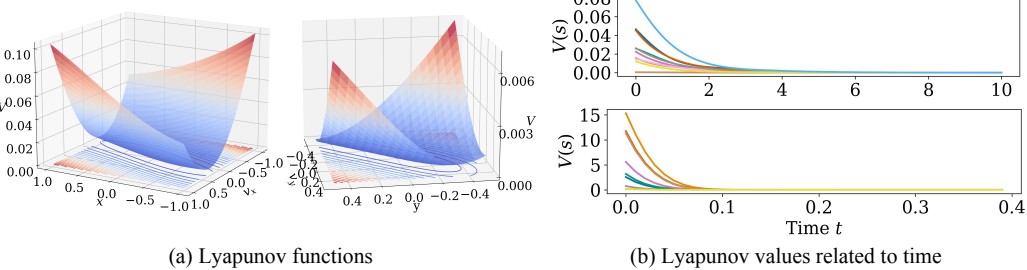

(a) Lyapunov functions   (b) Lyapunov values related to time

Figure 6: (a) The learnt CLF candidates are presented for the spacecraft (left) and 2D quadrotor (right) dynamics. (b) The CLF value along the simulated trajectories with random initial states in the spacecraft (up) and 2D quadrotor (down) dynamics, where values decreases monotonically as the system is driven to the equilibrium point.

## 6 CONCLUSION

We propose an end-to-end framework for learning Lyapunov functions and controllers for nonlinear dynamical systems. Our method reduces the constraints related to the Lyapunov conditions, simplifies the learning framework and decrease the difficulty in hyperparameter tuning by using a sum-of-squares neural network as the control Lyapunov function (CLF) and a CLF-based bounded nonlinear controller. Our approach exhibits excellent and robust performance in finding Lyapunov functions with the largest region of attraction (ROA) and the highest success rate using simple settings, highlighting its potential in better facilitating Lyapunov stability analysis and nonlinear control learning.

**Limitations and future works.** Our framework focuses on simplifying training process and improving the success rate, and further attention can be paid on the following aspects: 1. As the training complexity grows exponentially to the dimension of dynamics, it is essential to express dynamics in a data-efficient way, for example, achieving appropriate dynamics with reduced dimension. 2. To find a more powerful control policy with the stability guarantee, integrating our method with the model-based reinforcement learning framework will be further explored.

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
