# OpenReview forum: "Lyapunov Stability Learning with Nonlinear Control via Inductive Biases"
_ICLR.cc/2025/Conference — Submitted to ICLR 2025_

### Official Review · Reviewer_hMqL · 2024-10-23

**Soundness:** 2
**Presentation:** 1
**Contribution:** 1
**Rating:** 3
**Confidence:** 4

**Summary:**

This paper is concerned with synthesizing control Lyapunov functions via inductive biases. Authors propose a more simplified and relaxed conditions to achieve a better convergence rate, and easier verification. Finally, paper is concluded with two case studies (inverted pendulum and path following).

**Strengths:**

Authors propose more relaxed Lyapunov conditions by introducing inductive biases, which eliminates the difference of the learned
dynamics and the ground truth.

**Weaknesses:**

Paper is not mathematically rigorous. It is ambiguous whether authors contributed theoretically or not, since there appear to be no proof in the paper for the proposed claims. As an example, since neural networks are trained on finitely many points, how do you ensure that the proposed conditions are satisfied for the unseen data?

It also appears that the authors are not aware of the following paper: "Lyapunov-stable Neural Control for State and Output Feedback:
A Novel Formulation", ICML 2024, which is quite similar to this work. They modify Lyapunov condition to be $V(\zeta_t)\geq p$, as the verifier only needs to check the Lyapunov derivative condition when $\zeta_t$ is within the sublevel set $V(\zeta_t)< p$ (See Theorem 3.3).

Some key details have been left out, such as estimating Lipschitz constants of the learned dynamics, Lyapunov function and its corresponding controller, since these constants play a huge role in the verification process.

**Questions:**

1-You claim that previous encoding is complex and does converge globally, have you alleviated this problem theoretically?

2-Are you aware of the following paper: "Lyapunov-stable Neural Control for State and Output Feedback:
A Novel Formulation", ICML 2024? You have cited the previous work by these authors, but not the most recent one, which is quite similar to your method, with more theoretical contributions. How does your method compare to the aforementioned paper?

3- You have mentioned that the Lipschitz constant of the learned dynamics plays a role in the verification process, how do you obtain it? Trivial upper bound? If so, that will make your method quite conservative, since the trivial upper bound is orders of magnitude larger than the actual one (see "Efficient and accurate estimation of Lipschitz constants for deep neural networks", Neurips 2019).

4- It appears that you employ SMT solvers (mentioned in Algorithm I), which does not scale well if you have an over-parameterized network, which you claim that you have addressed. Have you eliminated the need for SMT solvers? since Lipschitz continuity is one way of providing guarantees.

5- I suggest you add a new subsection for notations and definitions, it helps with clarity and flow of the paper. Moreover, I personally think you need to motivate your problem better, since using just one library is not a valid research question; and using different libraries is not impractical (see line 63). If that is all your contribution, then this should have been a tool paper.

---

### Official Review · Reviewer_UN1T · 2024-10-24

**Soundness:** 2
**Presentation:** 3
**Contribution:** 2
**Rating:** 3
**Confidence:** 5

**Summary:**

The paper proposes an end-to-end framework for learning Lyapunov functions and controllers for nonlinear dynamical systems. The proposed method can reduce the constraints related to the Lyapunov conditions, simplify the learning framework, and decrease the difficulty in hyperparameter tuning by using a sum-of-squares neural network as the control Lyapunov function (CLF) and a CLF-based bounded nonlinear controller.

**Strengths:**

The paper is well-organized and easy to follow.

**Weaknesses:**

1. The literature review about the Lyapunov function for the stability analysis of nonlinear systems is insufficient. This topic is well-studied in the control community. The directions include deriving more relaxed stability conditions (compared with the ones in Eq. (3)) and constructing more advanced (e.g., piecewise, state-dependent, delayed) Lyapunov candidates. The paper should include comparisons with existing work, such as in [4].

2. The last paragraph on page 3: “To the best of our knowledge, there is no way to design a general form for the CLF and the bounded controller to completely satisfy all Lyapunov conditions.” This problem has also been well-studied recently in the RL community, such as in the work [1].

3. About  $\dot{\hat{s}} = f_{\phi_{1}}(s) +g_{\phi_{2}}(s)u$. The sensor samplings usually include system state $s$ and control command $u$. My question is how to use the sensor samplings to train MLPs to obtain $f_{\phi_{1}}$ and $g_{\phi_{2}}(s)$. Are they using the same sensor data? Are the training flow of the two MLP models the same? The paper should include such details. The paper shall have a section to include such details on obtaining $f_{\phi_{1}}$ and $g_{\phi_{2}}(s)$. Otherwise, I cannot conclude if the proposed method can be used in real systems.

4. The expression of $\dot{V}_{\phi}(s)$ on page 5 indicates the loss function needs the ground truth dynamics model, i.e., $f$ and $g$. This is very hard to achieve in practice. We can only approximate or learn them. Thus, the paper is expected to include studies or discussions about the influence of model error on the results.

5. All the experimental systems are numerical examples, so I can review the paper as a theory paper, as it does not include the implementation in real systems. However, the paper’s theoretical contributions are not sufficient. In my opinion, the most critical technical work can be represented by Eqs. (4)-(7), but they are direct extension and application of the results published in [2] and [3]. The expected contribution will be the theoretical support and explanations of the claimed features of the proposed framework, which, however, are not included in the paper.

6. The paper presents several strong and formal claims about the proposed approach, such as the claim on Page 6:  “Second, the verification of condition (3a) can be simply accomplished using deep learning packages” and  ``there is no overlooked region around the equilibrium mentioned in Equation (1) to pass the verification process.” I thought they were overstated because Eq. (3a) means the condition needs to hold for all the samples in the state domain. The closed domain can have an infinite number of system state samples. In reality, we can only have a vast number rather than infinite state samples. Formal theoretical support and explanations behind the claims are needed to address the comment.

7. The experiment section lacks comparisons with the work about the  Lyapunov functions for stability in the control community, such as the fuzzy Lyapunov function [4], and CLF in the RL community, such as the CLF-RL in [3].

References

[1] Westenbroek, T., Castaneda, F., Agrawal, A., Sastry, S., & Sreenath, K. (2022). Lyapunov design for robust and efficient robotic reinforcement learning. arXiv preprint arXiv:2208.06721.

[2] Grande, Davide, et al. "Augmented neural Lyapunov control." IEEE Access 11 (2023): 67979-67986.

[3] Liu, J., Fitzsimmons, M., Zhou, R., & Meng, Y. (2024). Formally Verified Physics-Informed Neural Control Lyapunov Functions. arXiv preprint arXiv:2409.20528.

[4] Fan, X., & Wang, Z. (2021). A fuzzy Lyapunov function method to stability analysis of fractional-order T–S fuzzy systems. IEEE Transactions on Fuzzy Systems, 30(7), 2769-2776.

**Questions:**

The critical questions arise from the $\dot{\hat{s}} = f_{\phi_{1}}(s) +g_{\phi_{2}}(s)u$. Sensor samplings usually include system state $s$ and control command $u$. How to use the sensor samplings to train MLPs to obtain $f_{\phi_{1}}$ and $g_{\phi_{2}}(s)$. Are they using the same sensor data? Are the training flow of the two MLP models the same? The paper should include such details.

---

### Official Review · Reviewer_N6FE · 2024-10-26

**Soundness:** 2
**Presentation:** 2
**Contribution:** 3
**Rating:** 5
**Confidence:** 4

**Summary:**

The paper proposes an end-to-end framework for learning Control Lyapunov Functions (CLFs) and controllers for nonlinear dynamical systems using inductive biases. By integrating Lyapunov conditions into neural network architecture, the framework enables stable, efficient learning with enhanced convergence rates and larger regions of attraction. Experiments validate its robustness across various control scenarios.

**Strengths:**

The framework achieves a larger region of attraction (ROA) compared to traditional approaches. Through continuous feedback from counterexamples, the model refines its stability properties, resulting in higher convergence rates and robust performance across a broader range of initial conditions in the state space.

**Weaknesses:**

1. Both Dai et al., 2021 and Chang et al. 2019 are discussed in detail in the paper.

Missing related work: Wu, Junlin, Andrew Clark, Yiannis Kantaros, and Yevgeniy Vorobeychik. "Neural lyapunov control for discrete-time systems." Advances in neural information processing systems 36 (2023): 2939-2955.

The setting is similar to Dai et al., 2021, except Dai et al., 2021 studies stability for NN dynamics, and this paper studies stability for nonlinear dynamics as in Chang et al. 2019.

2. The code is not attached.
3. the paper's presentation is a bit confusing. see questions below. If the authors can explain and make sense, i am willing to increase the score.

**Questions:**

"Second, the verification of condition (3a) can be simply accomplished using deep learning packages." What package, and how?

Can the authors provide code for the paper? I can't think of how to verify the Lyapunov conditions without using additional tools. In Algorithm 1, CheckSatisfiability(), what is this function exactly? It sounds like SMT solver.

How does the proposed approach verify stability near the origin? There must be numerical issues if you use SMT solver. It is hard to imagine there is no approximation near the origin, as is claimed in the paper.

---

### Official Review · Reviewer_zXvM · 2024-11-01

**Soundness:** 3
**Presentation:** 3
**Contribution:** 2
**Rating:** 5
**Confidence:** 5

**Summary:**

This paper introduces an end-to-end framework for learning CLF and controllers for nonlinear dynamical systems. By reducing constraints associated with Lyapunov conditions and simplifying the learning process, the proposed approach minimizes challenges related to hyperparameter tuning through the use of a sum-of-squares neural network as the CLF and a bounded nonlinear controller. The method demonstrates exceptional and consistent performance in identifying Lyapunov functions with the ROAs and highest success rates in straightforward settings.

**Strengths:**

The paper is well-written and easy to follow, with clear logic and a well-structured layout. It demonstrates a substantial amount of work, evident through the extensive experimental results and accompanying visualizations.

**Weaknesses:**

While the paper presents valuable insights, there are several areas that could be improved.
1. The discussion of related work is somewhat redundant, as the analysis of stability using Lyapunov theory and CLFs is well-known in the field and does not require extensive elaboration. Instead, the paper lacks sufficient citations and discussion regarding existing works that integrate learning and verification frameworks for CLF synthesis.
2. The paper's novelty is limited, which is evident not only in the use of SOS networks to fit the CLF and controller but also in its frequent reliance on existing work. First, while SOS networks are a common method for constructing polynomial invariants, they can easily fail to converge. Although the authors propose using a bounded controller learned via the tanh function to ensure convergence, this approach has been employed by many others and does not represent a significant innovation. Additionally, the requirement for the trainer to learn within a constrained range—similar to reinforcement learning for neural network controllers—does not add to its novelty. Second, comparing the proposed method to LQR techniques is not an ideal experimental setup. A more relevant comparison would be with contemporary learning and verification methods, such as those developed by Alessandro Abate at Oxford University and Fan ChuChu's work on CLFs.
References:《Neural Certificates for Safe Control Policies》《Formal Synthesis of Lyapunov Neural Networks》《 Fossil 2.0: Formal Certificate Synthesis for the Verification and Control of Dynamical Models》
3. Some experimental data presented in the main text may not hold significant value and could be better placed in an appendix. For instance, Table 2, Table 3, and Figure 6 do not substantially contribute to the core arguments of the paper and could distract from the main findings.
4. I noticed a lack of comparison regarding the time costs associated with the proposed method. Currently, the speed of synthesizing Lyapunov functions for low-dimensional nonlinear systems is significantly high, especially with learning methods based on SMT solvers and counterexample-guided approaches. These methods not only provide formal correctness guarantees (in contrast to the simulation tests used in this paper) but also utilize highly efficient neural network architectures that demonstrate strong learning capabilities.
5. There are several minor errors and suggestions for improvement: Line 231 contains an extra space; the notation for the 2-norm in Equation (2) should be reviewed; in Figure 3, the phrase "where black stars and red solid stars" is mentioned for the first time, but their overlap in the figure is unclear. It would be helpful to add a sentence explaining their overlap in the current figure.
6. I would like to see more examples involving high-dimensional systems, as this is one of the key advantages of the SOS method over other non-polynomial approaches. However, the examples provided in the experiments do not appear to be genuinely high-dimensional; for instance, the 6-dimensional example actually involves only 3 variables. Providing a few more complex examples (maybe something exists in the ARCH/ARCH-COMP repository) would better demonstrate the superiority of the proposed method.

**Questions:**

I would like to emphasize that this paper is well-written and clear. But there are some questions and uncertainties that I hope the authors can kindly address.
1. Why is a GPU used for tasks that could be effectively handled by a CPU? Based on the examples provided in the paper, it seems that a CPU would be sufficient for processing without the need for GPU acceleration. In my experience, even examples with 20 dimension can be effectively solved using a CPU. Providing a comparison of CPU vs GPU performance for your approach is more better.
2. I do not understand why the authors emphasize the reduction of constraints in CLF learning. Generally speaking, apart from the Lyapunov stability conditions, the other two conditions are relatively easy to satisfy. Moreover, the use of square nns to address these two conditions has already been established in prior work. Therefore, the challenge does not lie in solving the three conditions but rather in addressing the derivative condition.
3. In line 387, the authors use sampling methods to evaluate the quality of the learning results. However, this approach does not seem to provide an absolute guarantee of validity.

---

### Meta-Review · Area_Chair_C8jN · 2024-12-22

**Metareview:**

This paper studies an important topic of finding control Lyapunov functions to formally guarantee the stability of a controlled system. There have been many recent studies on applying deep learning based approaches to this problem and this research is very timely. The claimed novelty involves a design of a neural Lyapunov function that satisfies certain Lyapunov properties as inductive bias. However, reviewers have identified that a very similar formulation has been used in prior work, and the overall training framework is not novel. In addition, it lacks discussions or comparisons to some of the latest work in this field. Due to these key concerns, this paper cannot be accepted in its current form. The reviewers have all provided very constructive feedback for this paper, and I hope the authors can improve their work based on this feedback.

**Additional Comments On Reviewer Discussion:**

The authors did not provide a response. I've checked the reviews and they are all good quality.

---

### Decision · Program_Chairs · 2025-01-22

Reject